# Learning Exposure Correction in Dynamic Scenes

## ABSTRACT

Exposure correction aims to enhance visual data suffering from improper exposures, which can greatly improve satisfactory visual effects. However, previous methods mainly focus on the image modality, and the video counterpart is less explored in the literature. Directly applying prior image-based methods to videos results in temporal incoherence with low visual quality. Through thorough investigation, we find that the development of relevant communities is limited by the absence of a benchmark dataset. Therefore, in this paper, we construct the first real-world paired video dataset, including both underexposure and overexposure dynamic scenes. To achieve spatial alignment, we utilize two DSLR cameras and a beam splitter to simultaneously capture improper and normal exposure videos. Additionally, we propose an end-to-end Video Exposure Correction Network (VECNet), in which a dual-stream module is designed to deal with both underexposure and overexposure factors, enhancing the illumination based on Retinex theory. Experimental results based on various metrics and user studies demonstrate the significance of our dataset and the effectiveness of our method. The code and dataset will be available soon.

## CCS CONCEPTS

• **Computing methodologies** → **Machine learning**.

## KEYWORDS

video exposure correction, retinex theory, dataset, dynamic scene

## 1 INTRODUCTION

The images or videos taken across various scenarios may not always be ideal due to the changeable lighting conditions, which face underexposure and overexposure problems and yield unsatisfactory visual effects. How to improve inappropriately exposed visual data is gradually attracting the attention of researchers in the multimedia community. Due to the diversity of scenes and lighting conditions, the exposure correction process of visual data is particularly complicated, and overexposure and underexposure may exist at the same time (mixed exposure), which further enlarges the discrepancies of operations. Thus, many exposure correction methods have been proposed to handle this challenge problem automatically by utilizing deep learning technologies. Correcting underexposure and overexposure to normal exposures are much different from

**Unpublished working draft. Not for distribution.**

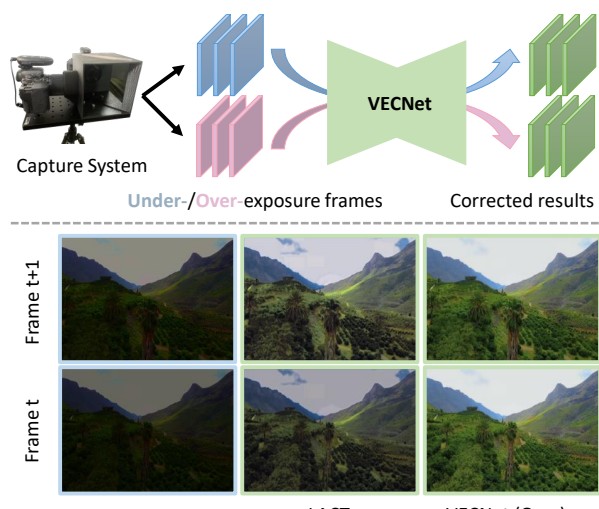

**Figure 1: The above sub-figure is the benchmark for our proposed video exposure correction. while the below sub-figures are the visual comparison between an image-based method LACT [2] and our proposed method. LACT takes single frames as input and results in temporal exposure incoherence with low visual quality. Our method utilizes temporal information to achieve consecutive exposures.**

each other, which leads to large discrepancies in training an image-based multi-exposure correction network. This challenge is further compounded when dealing with videos captured in dynamic scenes.

In recent years, fueled by various paired image exposure dataset [1, 3], several methods [12, 13, 26, 32, 36] that focus on learning image exposure correction have been proposed. However, these methods generally do not handle videos of dynamic scenes, as directly applying them in a frame-by-frame manner often results in inter-frame exposure inconsistency with low visual quality [16, 20], as shown in the below sub-figures of Fig. 1. Some methods [9, 27, 34] focus on enhancing underexposed videos taken from low-light dynamic scenes. However, these methods can only achieve a single exposure correction. Due to the discrepant representations between underexposure and overexposure, the correction procedures differ greatly from each other, as shown in Fig. 3(a). Training a mixture of multi-exposure data in their models often leads to poor performance across different exposure levels, rendering these methods unsuitable for practical applications.

To overcome the above challenges, in this work, we focus on the multi-exposure correction presented in the real-world videos for the first time. As of our current understanding, there is a noticeable absence of well-studied works or benchmark datasets in this domain. On the one hand, we aim to collect a high-quality video dataset with both underexposure and overexposure, and the main difficulties are as follows. First, collecting paired video datasets requires capturing

two videos, one under abnormal exposure conditions and another under normal light of the same dynamic scene and camera motion. Second, it requires accurate alignment of each pair of corresponding frames in the two videos in both spatial and temporal dimensions. To address the above issues, we first design a two-camera system with a beam splitter to ensure no parallax between the two cameras with different aperture and ISO parameter settings. Two DSLR cameras simultaneously capture improper and normal exposure videos. Then we perform precise alignment on the captured pairs to make them aligned with each other. Furthermore, the well-exposed videos are manually rendered by professional photographers and then serve as the ground truth of the dataset. Finally, we construct a new dataset of 119 high-quality videos named **DIME**, which stands for "**D**ynamic scenes **I**n **M**ultiple **E**xposure". It contains diverse real-world scenes, camera and object motions, and each paired video is accurately aligned along the spatial and temporal dimensions.

On the other hand, since there are varying degrees of improper exposure in the constructed dataset, existing methods prove inadequate for deployment. Thus, on the other hand, we propose an end-to-end **V**ideo **E**xposure **C**orrection **Net**work (**VECNet**) to enhance the videos with multiple and improper exposures. Specifically, to adaptively learn the overexposed and underexposed representations, we formulate a dual-stream strategy to enhance various illumination components based on Retinex theory. For complementary reflectance component learning, we design a novel multi-frame alignment module that aligns neighboring frames into the middle one to effectively obtain comprehensive feature representations. In brief, our contributions can be summarized as follows:

- We construct the first high-quality paired video exposure correction dataset for dynamic real-world scenes with multiple exposures, camera and object motions, and precise spatial alignment.
- We propose an effective exposure correction network based on Retinex theory to enhance overexposed and underexposed videos.
- We conduct extensive experiments to demonstrate the superiority of our dataset and method.

## 2 OUR DIME DATASET

To learn video exposure correction tasks by training our model, we need a large number of videos, including realistic overexposure and underexposure errors and corresponding paired and properly exposed ground truth videos. As discussed in Sec. 2, such datasets are currently not publicly available to support video exposure correction research. Therefore, our first task is to create a novel dataset that captures various videos in dynamic and real-world scenes.

### 2.1 Hardware Design

Capturing multi-exposure image and video pairs for static scenes can be easily realized with a single-exposure adjustable DSLR camera [9] or an electric slide rail [27], which make it impractical to capture paired videos in wider dynamic scenes. We consider utilizing two cameras with different aperture and ISO parameter settings. To keep the cameras in sync, we use an infrared remote control to signal two camera-mounted receivers for simultaneous capture. To prevent parallax problems caused by different shooting positions

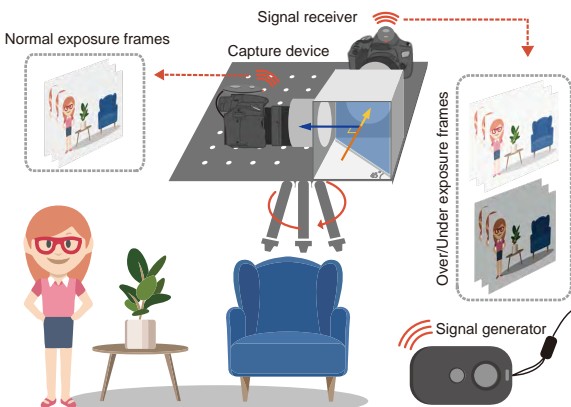

**Figure 2: We built the optical system to capture the over-/under- and normal exposure video pairs.**

and ensure the same motion trajectory, inspired by [10, 27], we use a beam splitter to split the incident light into two beams with a ratio of 1:1 and enter the two camera lenses, respectively, as shown in Fig. 2. In order to receive the natural incident light from the same viewpoint, we design and print an opaque 3D model box, which is also used to hold the beam splitter and connect the cameras. These components are concentrated on an optical breadboard and then fixed to the tripod to improve stability.

### 2.2 Data Collection

We use two Canon EOS R10 cameras to collect paired videos. Specifically, to minimize the loss of detail in shadows and highlights, we set one camera to automatic exposure mode and capture video in Canon Log[1] format, which retains more visual information throughout the dynamic range. We then hand it over to professional photographers using DaVinci Resolve Studio[2] software for manual rendering, ultimately producing high-quality and normal-exposure videos. The other camera captures low-quality abnormal exposure videos with the commonly used sRGB format. All the other settings are set to default values to simulate real capture scenarios. We collect paired 10-second clips for each dynamic scene.

### 2.3 Data Processing

Due to the slight errors in manually assembling the above hardware, there is still a misalignment between the paired videos. Therefore, we utilize a two-stage frame alignment strategy to obtain aligned pairs. First, we estimate a homography matrix between the under-/overexposed frame and the corresponding normal exposed frame using the matched SIFT [19] key points. In this way, we can roughly crop matching regions into corresponding frames. Then we utilize a traditional flow estimation algorithm called DeepFlow [35] to perform pixel-wise alignment. Finally, we use center cropping to remove alignment artifacts around boundaries, producing precise spatially aligned frames. Note that we only perform alignment correction on normal exposed frames and make the low-quality input

---

[1]https://cam.start.canon/en/C004/manual/html/UG-03_Shooting-2_0080.html
[2]https://www.blackmagicdesign.com/products/davinciresolve/studio

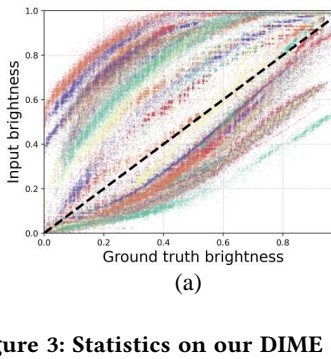
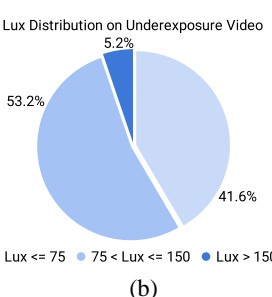
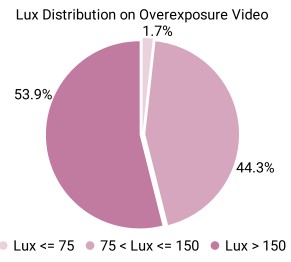
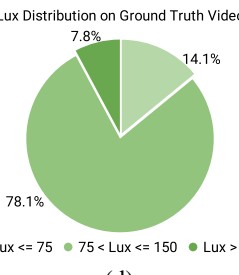

(a)                                    (b)                                    (c)                                    (d)

**Figure 3: Statistics on our DIME dataset. (a) is the input-ground truth brightness mapping curve statistics. (b) Luminance distribution for overexposure videos. (c) Luminance distribution for underexposure videos. (d) Luminance distribution for ground truth videos.**

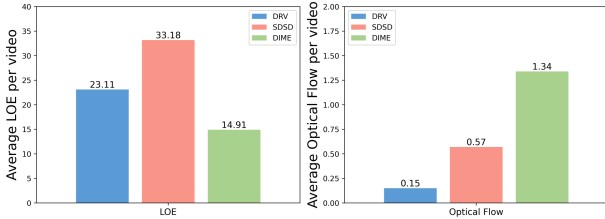

**Figure 4: LOE and optical flow of different datasets.**

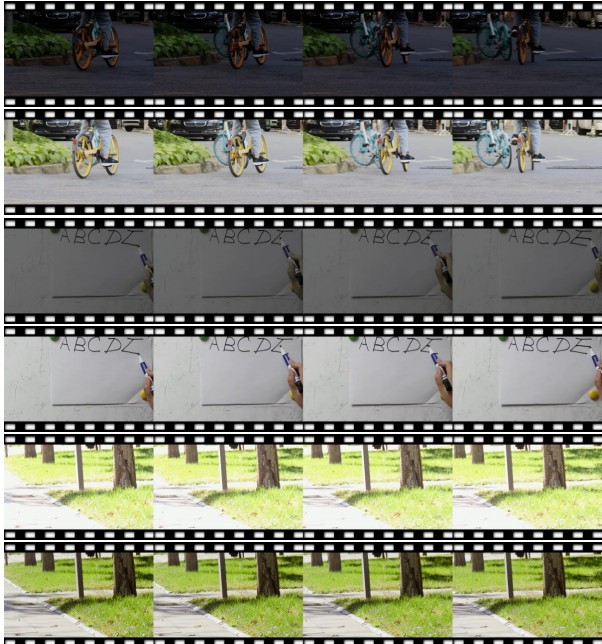

**Figure 5: Example videos from our DIME dataset cover under-/over-exposures, indoor/outdoor, and camera/object motions.**

of our network consistent with real captured abnormal exposed frames. Additionally, we introduce lightness-order-error (LOE) [29] and optical flow [7] to quantitatively evaluate alignment performance and motion activity. As shown in Fig. 4, the paired frames of our dataset are better aligned and have larger movement than that of the low-light enhancement datasets, including DRV [5] and SDSD [27], indicating stronger performance in real dynamic scenes.

We totally captured 119 groups of videos with a total of 21,951 frames to form our DIME dataset. Each video consists of 100-200 frames, and the resolution is $1,920 \times 1,080$. Fig. 3 presents some statistical analysis results. Fig. 5 gives three examples of our aligned video pairs under under-/over-exposure and normal conditions. Our captured videos vary from indoor to outdoor in real scenes and include camera and object motion types. See more detailed information in the supplementary materials.

## 3 METHODOLOGY

### 3.1 Overview

The Video Exposure Correction (VEC) task can be formulated as seeking a mapping function $\mathcal{F}$, which maps consecutive 8-bit per channel sRGB frames $I$ to enhanced frames $O$ such that $O = \mathcal{F}(I)$. During the training step, given 2N+1 consecutive frames $I_{[t-N:t+N]}$ captured under different exposure conditions, the center frame is denoted as the reference frame while the others are supporting frames. The proposed end-to-end VECNet aims to restore the exposure of the reference frame $I_t$ with the help of supporting frames, thereby generating the output frame $O_t$. Based on Retinex theory [15], the overall architecture of VECNet is shown in Fig. 6, which is composed of three sub-networks: Multi-frame Fourier Alignment Module (MFA), Dual-stream Illumination Construction

Unit (DIC) and Two-stage Synthesis Restoration Unit (TSR). The MFA module first aligns supporting frames with the reference frame to maintain temporal consistency and then learns the reflectance map together. Next, the DIC unit estimates dual-path illumination maps for the reference frame to adaptively adjust its underexposure and overexposure factors. Finally, the TSR unit fuses the above illumination and reflectance maps at feature and image levels, obtaining the high-quality output frame $O_t$ with proper exposure.

### 3.2 Multi-frame Fourier Alignment

Directly applying image exposure correction [1, 12] to each frame can cause flickering [27] due to ignoring temporal information. To take advantage of the multi-frame information, we take consecutive frames as the model's input. However, since the texture information

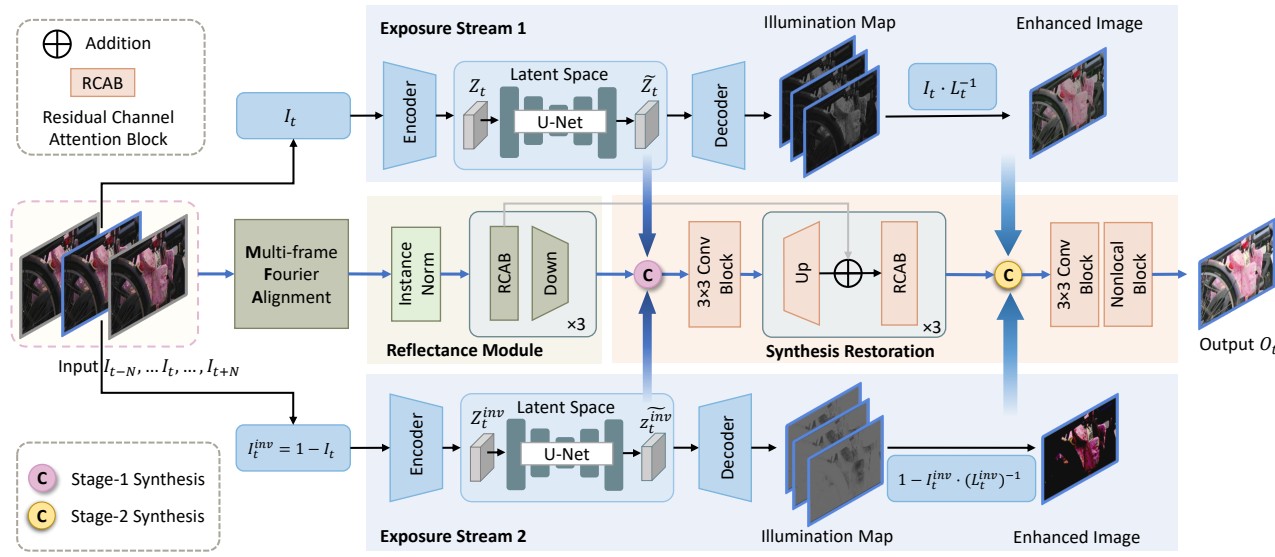

**Figure 6: Overview of our framework. It contains three modules, including Multi-frame Fourier Alignment (MFA), Dual-stream Illumination Construction (DIC), and Two-stage Synthesis Restoration (TSR).**

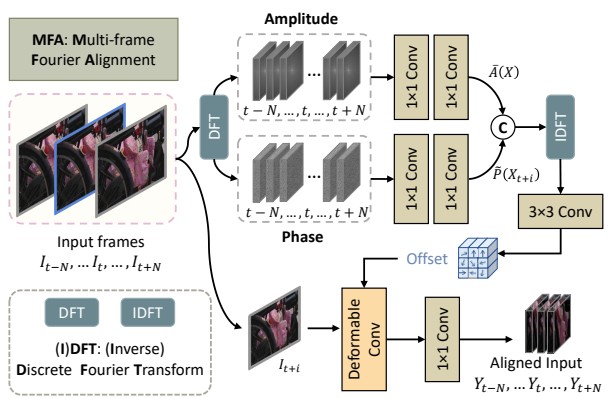

**Figure 7: The details of the Multi-frame Fourier Alignment module. We apply the discrete Fourier transformation to map the frames from pixel space to Fourier space, then adopt the deformable convolution for alignment.**

of different frames is misaligned, a direct fusion of multiple frames could cause blur and artifacts [23, 27]. Existing works mainly warp supporting frames to the reference frame by pyramid cascading and deformable convolution [30, 37] or group shift operations [17] directly at the feature level. However, in the video exposure correction task, there are differences in degraded exposure between frames as camera motion produces different light positions. Such exposure mutation interferes with alignment. Recently, [12] conducts exposure correction from a Fourier-based perspective, i.e., the amplitude component of an image reflects the lightness representation, while the phase component corresponds to structures and is less related to lightness. Therefore, we reduce the influence of lightness factors on alignment by adapting the amplitude component. To this

end, we take multi-frame alignment using the Fourier transform in this section, as shown in Fig. 7.

Specifically, given a channel in a single frame $x \in \mathbb{R}^{H \times W}$, the Fourier transform $\mathcal{F}$ converts $x$ to a complex component $X$ in Fourier space, which can be written as:

$$\mathcal{F}(x) = \frac{1}{\sqrt{HW}} \sum_{h=0}^{H-1} \sum_{w=0}^{W-1} x(h,w) e^{-j2\pi(\frac{h}{H}u + \frac{w}{W}v)}, \quad (1)$$

Then, amplitude $\mathcal{A}(X)$ and phase $\mathcal{P}(X)$ components can be divided from the complex component $X$:

$$\mathcal{A}(X) = \sqrt{R^2(X) + I^2(X)},$$
$$\mathcal{P}(X) = arctan \frac{I(X)}{R(X)}, \quad (2)$$

where $R$ and $I$ denote the real and imaginary parts of the complex $X$. We also apply the same strategy to each channel of input frames and then get consecutive amplitude components $\mathcal{A}(X_{[t-N:t+N]})$ and phase components $\mathcal{P}(X_{[t-N:t+N]})$. Since the amplitude component responds more to the lightness, we obtain $\bar{\mathcal{A}}(X)$ by aggregating multi-frame amplitude to achieve a unified exposure representation:

$$\bar{\mathcal{A}}(X) = \mathcal{H}[\mathcal{A}(X_{t-N}) \odot ... \mathcal{A}(X_{t+N})], \quad (3)$$

where $\mathcal{H}$ is the aggregating function composed of several convolution and relu function layers. Then the phase component $\tilde{\mathcal{P}}(X_{t+i})$ of the supporting frames and aggregated amplitude component $\bar{\mathcal{A}}(X)$ are recombined by the inverse Fourier transform $\mathcal{F}^{-1}$. The process is formulated as:

$$\tilde{x}_{t+i} = \mathcal{F}^{-1}(\bar{\mathcal{A}}(X), \tilde{\mathcal{P}}(X_{t+i})), \quad (4)$$

Then the supporting frames are warped to those of the reference frame by the deformable convolution network (DCN) [6]. We compute the offset $\delta x_{t+i}, i \in [-N, N], i \neq 0$ learned from corresponding

$\tilde{x}_{t+i}$) and $\tilde{x}_t$) for the DCN. The aligned phase component $\tilde{\mathcal{P}}(Y_t)$ is obtained with the learned offset. The process can be formulated as:

$$
\begin{aligned}
\delta x_{t+i} &= \mathcal{M}_i(\tilde{x}_{t+i} \odot \tilde{x}_t), \\
Y_{t+i} &= \mathcal{G}_i(DCN(\tilde{x}_{t+i}, \delta x_{t+i})),
\end{aligned}
\tag{5}
$$

where $\odot$ denotes channel concatenation, and $\mathcal{M}_i$ and $\mathcal{G}_i$ are the mapping functions composed of several convolution layers. Then the aligned features consisting of multiple single-channel $Y_{t+i}$ are sent to learn the reflectance map.

## 3.3 Dual-stream Illumination Construction

To learn adaptive representations for both underexposure and overexposure situations, we exploit dual-stream illumination construction in our model. From the Retinex theory, the reference frame $\mathcal{I}_t$ can be typically decomposed into an illumination map $L$ and a reflectance map $\mathcal{R}$. However, this strategy can only be learned from underexposure conditions, as $L$ falls into the range of values within $[0, 1]$. Existing Retinex-based methods [4, 8] would increase the exposure of the overexposed inputs.

To suppress overexposure while enhancing underexposure, we propose a dual-stream mechanism to achieve illumination construction, sharing parameters with each other. Specifically, as shown in Fig. 6, we treat the antithetic exposure $\mathcal{I}_t^{inv}$ as the reverse frame of $\mathcal{I}_t$. We first encode $\{\mathcal{I}_t, \mathcal{I}_t^{inv}\}$ to $z_t, z_t^{inv}$ in the latent space. Then we use a U-Net [22] to learn the mapping from $\{z_t, z_t^{inv}\}$ to $\{\tilde{z}_t, \tilde{z}_t^{inv}\}$. Finally, we decode $\{\tilde{z}_t, \tilde{z}_t^{inv}\}$ to $\{L_t, L_t^{inv}\}$ in the image space. By calculating with the learned and extended illumination map $L^{inv}$, we can obtain the enhanced image of overexposure $\mathcal{I}_t^o$. This process can be formulated as:

$$
\begin{aligned}
\mathcal{I}_t^u &= \mathcal{I}_t \cdot L_t^{-1}, \\
\mathcal{I}_t^{inv} &= 1 - \mathcal{I}_t, \\
\mathcal{I}_t^o &= 1 - \mathcal{I}_t^{inv} \cdot (L_t^{inv})^{-1},
\end{aligned}
\tag{6}
$$

where $\mathcal{I}_t^u$ represents the enhanced frame of underexposure, and $\cdot$ denotes element-wise multiplication.

## 3.4 Two-stage Synthesis Restoration

We apply a reflectance sub-network $\mathcal{T}$ to learn the mapping from aligned consecutive frames $\mathcal{Y}_{[t-N:t+N]}$ to the reflectance map, each frame $\mathcal{Y}_t$ is composed of $\{\tilde{x}_t^R, \tilde{x}_t^G, \tilde{x}_t^B\}$. This sub-network is combined with instance normalization [24] (IN) operation and a series of residual channel attention blocks (RCAB) [38] and downsampling operations. The IN operation is used to map different exposure features to exposure-invariant feature space [11] while RCABs are presented to learn high-level residual feature information. In the synthesis process of stage one, the fusion network $\mathcal{S}_1$ takes separately learned dual-illumination $\{\tilde{z}_t, \tilde{z}_t^{inv}\}$ and reflectance map to synthesize intermediate enhanced feature $\tilde{I}_t$. It contains a series of residual channel attention blocks and upsampling operations. In addition, skip connections are applied for the corresponding layers of the reflectance module and the synthesis module.

In the synthesis process of stage two, the fusion network $\mathcal{S}_2$ takes two enhanced images $\{\mathcal{I}_t^u, \mathcal{I}_t^o\}$ and the intermediate feature $\tilde{I}_t$ as inputs to regress the final enhanced image $O_t$. It uses several combining modules of convolution layers and non-local blocks [31]

to generate a three-channel weight map, and then multiplies the above three inputs by their respective weights by channel to obtain the final result $O_t$. The process can be formulated as:

$$
\begin{aligned}
\tilde{I}_t &= \mathcal{S}_1(\tilde{z}_t \odot \tilde{z}_t^{inv} \odot \mathcal{T}(\mathcal{Y}_{[t-N:t+N]})) \\
O_t &= \mathcal{S}_2(\mathcal{I}_t^u, \mathcal{I}_t^o, \tilde{I}_t)
\end{aligned}
\tag{7}
$$

## 3.5 Objectives

We incorporate a range of loss terms to facilitate the training of our model. We employ the commonly utilized pixel-wise charbonnier loss term, denoted as $\mathcal{L}_{pix}$, to measure the accuracy of pixel reconstruction. To guide the network to estimate and smooth the dual illumination maps, we apply the total variation [28] loss term $\mathcal{L}_{tv}$. They can be formulated as:

$$
\mathcal{L}_{tv} = \sum_c [(\partial_{x/y} L_t)^2 + (\partial_{x/y} L_t^{inv})^2],
\tag{8}
$$

where all channels (c) of all pixels are summed, $\partial_{x/y}$ are partial derivatives in horizontal and vertical directions in the image space. In addition, to ensure exposure continuity between video frames, we introduce the amplitude consistency loss term $\mathcal{L}_{amp}$, which can be formulated as:

$$
\mathcal{L}_{amp} = ||\mathcal{A}(X_{O_t}) - \mathcal{A}(X_{I_t^{gt}})||_F^1,
\tag{9}
$$

where $X_{(\cdot)}$ represents the frequency component of output $O_t$ or ground truth normal exposed image $I_t^{gt}$ in Fourier space. The overall function can be written as:

$$
\mathcal{L}_{total} = \lambda_1 \mathcal{L}_{pix} + \lambda_2 \mathcal{L}_{tv} + \lambda_3 \mathcal{L}_{amp},
\tag{10}
$$

where $\lambda_1$, $\lambda_2$, and $\lambda_3$ are weight hyperparameters.

# 4 EXPERIMENTS

## 4.1 Implementation Details

We implement our framework with Pytorch on a single NVIDIA GeForce RTX 3090 GPU. We use $N = 2$ consecutive frames without intervals as input. We divide the DIME dataset into training, validation, test sets according to the number of videos in 90:9:20. Then we use the training set to train our model with a total of 400,000 iterations. The parameters of the network are optimized by the ADAM optimizer with $\beta_1 = 0.9$ and $\beta_2 = 0.99$. The learning rate is 0.001, while the batch size is 8. The frames apply random horizontal and vertical flipping to augment the input data. We train on patches of size $256 \times 256$. The weights for the terms in the loss function in Eq. (9) are $\lambda_1, \lambda_2, \lambda_3 = 1.0, 0.01, 100.0$. The additional experimental results are presented in the supplementary materials.

## 4.2 Baselines

Since there is currently no related method for video exposure correction, we adopt several image exposure correction methods for comparison, including MSEC [1], DRBN-ENC [11], ECLNet [13], FECNet [12], and LACT [2]. We also take low-level tasks adjacent to video exposure correction, including several methods in the fields of video low-light enhancement and video restoration, such as SMOID [14], SDSD [27], RVRT [18], and DIDNet [9].

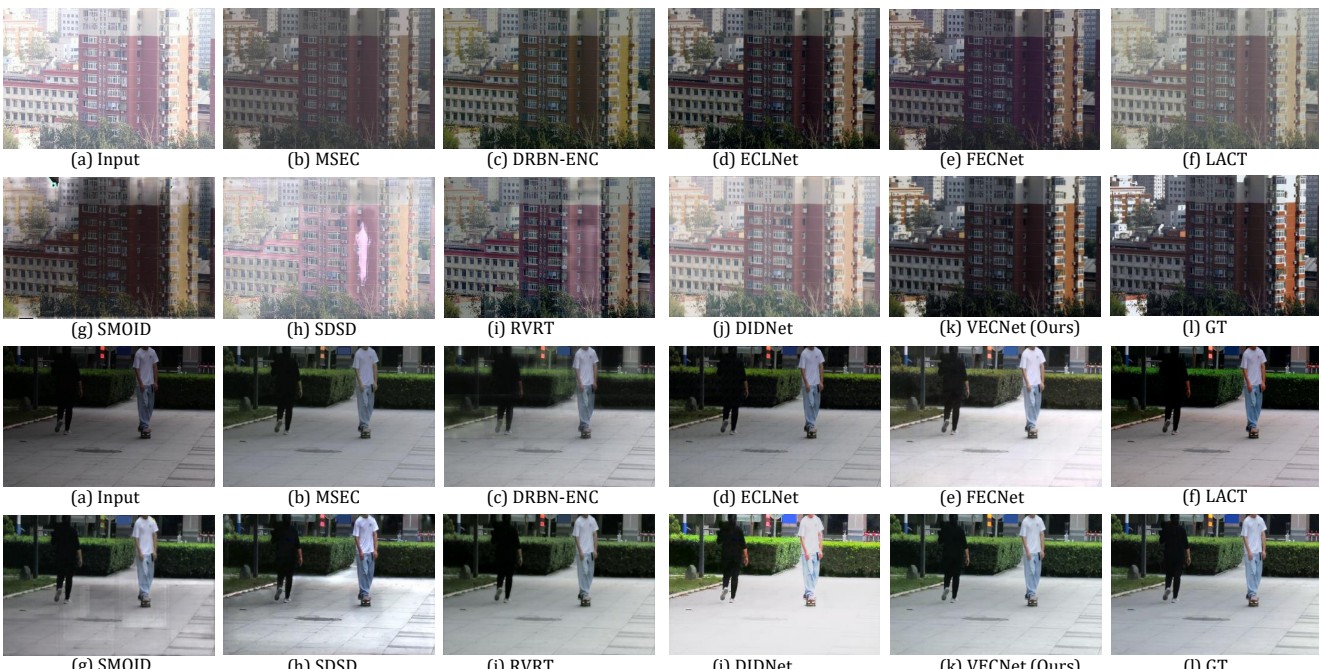

Figure 8: Qualitative comparisons of the multi-frame results with different methods on the DIME dataset.

Table 1: Quantitative PSNR and SSIM results of different methods on Underexposed and Overexposed samples of the DIME dataset. A higher metric indicates better performance. The best and second results are marked in bold and underlined, respectively.

| Method | Under PSNR | Under SSIM | Over PSNR | Over SSIM | Average PSNR | Average SSIM |
|---|---|---|---|---|---|---|
| SMOID [14] | 16.78 | 0.8037 | 17.65 | 0.8675 | 17.22 | 0.8356 |
| SDSDNet [27] | 17.51 | 0.8511 | 18.83 | 0.8720 | 18.17 | 0.8616 |
| RVRT [18] | 16.62 | 0.7859 | 16.93 | 0.8555 | 16.78 | 0.8207 |
| DIDNet [9] | 17.89 | 0.8587 | 21.01 | 0.9022 | 19.45 | 0.8805 |
| MSEC [1] | 16.22 | 0.8271 | 18.01 | 0.8854 | 17.12 | 0.8563 |
| DRBN-ENC [11] | 17.69 | 0.8580 | 20.12 | 0.9203 | 18.91 | 0.8892 |
| ECLNet [13] | 16.80 | 0.8062 | 19.56 | 0.8915 | 18.18 | 0.8489 |
| FECNet [12] | 17.61 | 0.8601 | 21.02 | 0.9109 | 19.32 | 0.8855 |
| LACT [2] | 18.01 | 0.8651 | 20.36 | 0.9153 | 19.19 | 0.8902 |
| VECNet (Ours) | **18.18** | **0.8687** | **22.04** | **0.9345** | **20.11** | **0.9016** |

Table 2: Quantitative results of different methods in terms of NIQE and ALV. A lower metric indicates better performance. The best and second results are marked in bold and underlined, respectively.

| Method (Publication) | DIME NIQE | DIME ALV | LLIV-Phone NIQE | LLIV-Phone ALV |
|---|---|---|---|---|
| SMOID [14] (ICCV'19) | 5.43 | 35.32 | 6.76 | 6.06 |
| SDSDNet [27] (ICCV'21) | 4.35 | 18.04 | 4.07 | 8.99 |
| RVRT [18] (NeurIPS'22) | 4.17 | 20.81 | 5.75 | 7.07 |
| DIDNet [9] (ICCV'23) | 7.25 | 42.60 | 6.93 | 10.32 |
| MSEC [1] (CVPR'22) | 5.38 | 37.28 | 5.80 | 232.8 |
| DRBN-ENC [11] (CVPR'22) | 4.48 | 45.51 | 5.24 | 129.4 |
| ECLNet [13] (ACMMM'22) | 4.40 | 23.31 | 5.38 | 141.0 |
| FECNet [12] (ECCV'22) | 4.52 | 29.11 | 5.26 | 105.5 |
| LACT [2] (ICCV'23) | 5.01 | 35.57 | 5.62 | 171.6 |
| VECNet (Ours) | **4.08** | **15.65** | **4.02** | **4.94** |

## 4.3 Quantitative Evaluation

We demonstrate the superiority of our method and the impact of the DIME dataset through experiments in this section. We conduct quantitative experiments on the test set, which consists of 10 underexposed video pairs and 10 overexposed video pairs with a total of 3,824 frames, as described in Sec. 2. We adopt the following three standard metrics to evaluate the pixel-wise accuracy and perceptual quality of our results: reference-based (i) Peak Signal-to-Noise Ratio

(PSNR) and (ii) Structural Similarity Index Measure (SSIM) [33], non-reference-based (iii) Natural Image Quality Evaluator (NIQE) [21] and (iv) Average Luminance Variance (ALV) [16] metrics. Table 1 reports the quantitative evaluation results of different methods. As exhibited in the table, our proposed VECNet outperforms all other methods in PSNR and SSIM metrics, demonstrating its superior performance in video exposure correction tasks.

**Validation on unpaired videos.** To evaluate the generalization ability of our method under challenging dynamic scenes, we also

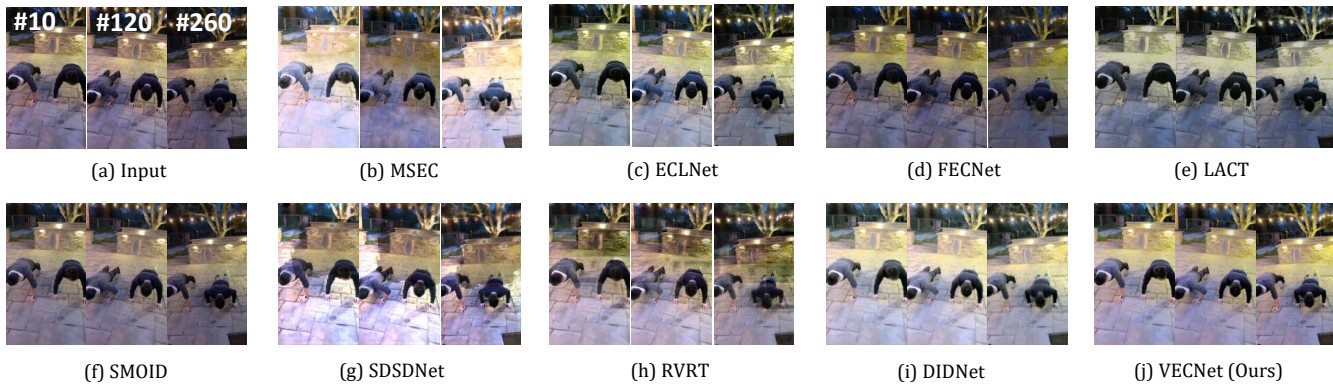

Figure 9: Qualitative comparisons of the multi-frame results with different methods on the LLIV-Phone dataset.

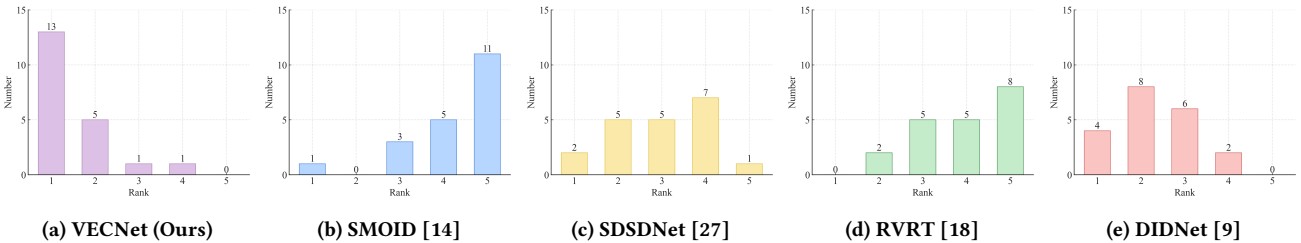

Figure 10: The results of five methods in the user study. In each histogram, the x-axis denotes the ranking index (1∼5, 1 represents the highest), and the y-axis denotes the number of images in each ranking index.

Table 3: Quantitative results of different methods for multi-exposure videos captured from various devices and the internet, evaluated by NIQE and ALV metrics.

| Method | iPhone | XiaoMi | Internet | Average |
|---|---|---|---|---|
| LACT [2] | 4.42/38.32 | 4.85/47.50 | 4.11/24.13 | 4.46/36.65 |
| DIDNet [9] | 4.10/30.90 | 4.55/27.66 | 3.72/18.72 | 4.12/25.76 |
| VECNet | 3.28/22.95 | 4.40/26.04 | 3.45/10.77 | **3.71**/**19.92** |

conduct extensive experiments on the LLIV-Phone dataset [16]. Since there is no ground truth, all compared baselines are only tested on the dataset. Note that we report the results of our model trained on the DIME training set without further tuning or retraining on any of these datasets. Table 2 reports the quantitative results. Our method achieves the best results in terms of NIQE and ALV, demonstrating stability when handling real-world videos with various motions.

**Various kinds of device video enhancement.** We collect a total of 30 abnormal exposure videos from three devices, 10 from each device. We test them with the compared methods pretrained on the DIME training set. As described in Table 3, our method achieves better results, displaying its effectiveness for more applications. In addition, we present the luminance curves of the enhanced video frames to verify the degree and continuity of the exposure, as shown in Fig. 12. It can be seen that our VECNet behaves more

stable, especially in some cases with drastic light changes (solid line).

## 4.4 Qualitative Evaluation

We perform thorough qualitative evaluations on the DIME and LLIV-Phone datasets to assess the performance of our proposed method. We present single-frame and multi-frame results in Fig. 8 and Fig. 9, which indicates that VECNet delivers more natural and reasonable enhancement. In particular, low-light video enhancement methods mainly produce frames with unpleasing regions, leading to substantial loss of texture. Image-based exposure correction methods result in significant color deviation, which adversely affects the visual quality of the images. SMOID and RVRT produce frames with severe blocking artifacts. Moreover, image-based methods generate inconsistent exposure frames as the temporal information has not been well utilized to avoid flickering. After a comprehensive evaluation of the comparative results of different methods on two datasets, our proposed method exhibits excellent visual performance in terms of global brightness, color recovery, and detail while maintaining temporal stability without flickering artifacts and motion blur.

## 4.5 User Study

We conduct a user study with 20 participants to evaluate the subjective perceptions of different methods. We select 20 testing videos from the DIME dataset. The videos are then enhanced using 5 video enhancement methods (SMOID, SDSDNet, RVRT, DIDNet,

**Table 4: Ablation study for investigating the components of the specific modules. For the MFA module, we ablate the alignment with Fourier transform. For the DIC module, we ablate the single and dual streams.**

| Model | MFA | DIC | | TSR | | PSNR | SSIM |
|-------|-----|-----|----|-----|-----|------|------|
| | Fourier | single | dual | stage-1 | stage-2 | | |
| (a) | | ✓ | | ✓ | | 19.03 | 0.8838 |
| (b) | ✓ | ✓ | | ✓ | | 19.44 | 0.8905 |
| (c) | ✓ | | ✓ | ✓ | | 19.86 | 0.8972 |
| (d) | ✓ | | ✓ | ✓ | ✓ | **20.11** | **0.9016** |

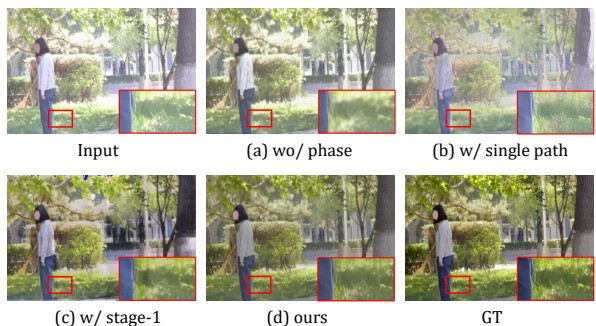

**Figure 11: Visualizations of the ablation study on the proposed modules.**

and VECNet) to perform pairwise comparisons. For a fair comparison, we provide the original input videos and present 5 kinds of enhancement results, randomly changing the display order to prevent bias. The user study is conducted in the same environment (room, display, and light). Participants are asked to rate their video quality from best to worst. The human subjects are instructed to consider authenticity, exposure artifacts, texture contrast, realistic color, and temporal stability. We calculate the average ranking of each method on each video and rank the results. As a result, each method is assigned a rank of 1–5 on that video.

The final results are shown in Fig. 10. VECNet has 13 results ranked 1, 5 results ranked 2, and a few results ranked 3-5 out of the 20 videos evaluated. When the histograms are compared, it is clear that our methods produce better results for human subjective evaluations across all baselines.

### 4.6 Ablation Study and Analysis

We provide a series of ablation studies to evaluate the effectiveness of each component of the proposed method. The experiments are conducted on the DIME dataset. The ablation results are shown in Table 4, Fig. 11, and supplementary materials.

For the MFA module, we remove it and directly fuse the original unaligned multiple frames to learn reflection maps to generate the final result. It can be seen that MFA brings expected quantitative performance improvements. The results of Fig. 11 (b) are more visually appealing with fewer noises and motion blurs than (a), which further demonstrates the effectiveness of MFA. For the DIC unit, we replace the symmetrical exposure stream with a single

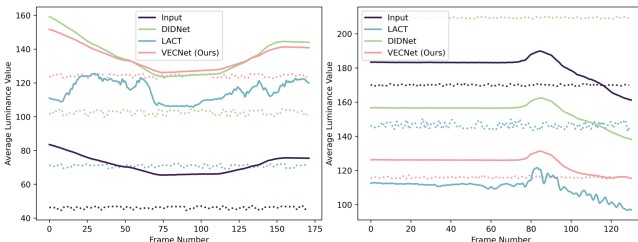

**Figure 12: The lux curves of the video are enhanced by different methods. Left are underexposure cases, while right are overexposure cases.**

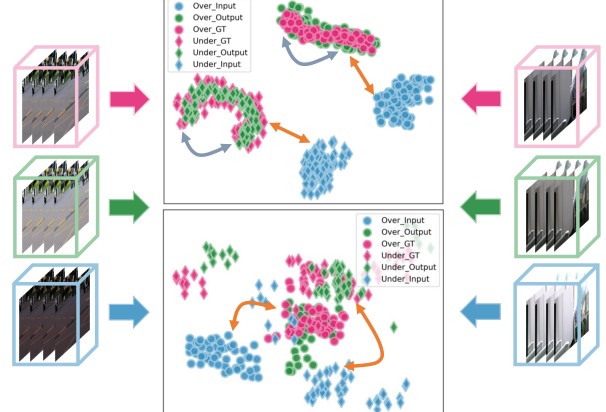

**Figure 13: t-SNE [25] visualization when taking a testing video (upper part) and testing set (lower part) for example.**

path setting. It can be seen that the overexposure correction shows little improvement from Fig. 11 (b). For the TSR unit, we split the TSR unit into two separate stages. Comparing Fig. 11 (c) and (d), we can see that our two-stage strategy helps preserve more color and texture information.

In addition, we present the statistical visualization of the results in the feature space. As shown in Fig. 13, after being processed by our method, the underexposure and overexposure representations tend to be intersected together with the corresponding ground truth. It demonstrates the effectiveness of our method for correcting their exposure representations.

### 5 CONCLUSION

We build the first high-quality paired video exposure correction dataset for dynamic real-world scenes with multi-exposures, camera and object motions, and precise spatial alignment. A benchmark dataset is provided for both training and evaluation of video exposure correction methods. Based on the dataset, we propose a method for dealing with the underexposed and overexposed inputs in a dual-stream manner. By utilizing phase alignment and synthesis modules, the exposure of the videos is well corrected and restored. Experiments demonstrate that the proposed method outperforms several image and video methods in low-level tasks adjacent to video exposure correction.

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
