# OpenReview forum: "Learning Exposure Correction in Dynamic Scenes"
_acmmm.org/ACMMM/2024/Conference — MM2024 Oral_

### Official Review · Reviewer_ZN1W · 2024-05-14

**Rating:** 4
**Confidence:** 3

**Summary:**

The goal of this work is to solve the problem of video exposure correction. To achieve this goal, this work proposes the first real-world paired video dataset, which includes both underexposure and overexposure dynamic scenes. At the same time, this work proposes a video exposure correction network to solve the underexposure and overexposure problems through two exposure streams.

**Strengths:**

1. This work proposes the first real-world video exposure correction dataset, including both underexposure and overexposure dynamic scenes.
2. This work proposes an end-to-end video exposure correction network to solve the underexposure and overexposure problems through two exposure streams.
3. This work conducts extensive experiments to demonstrate the effectiveness of their approach.

**Limitations:**

1. It may be better to give some comparisons with previous methods on the model's inference overhead, especially the model's calculation amount and inference time.
2. Giving more comparisons with previous data sets may better illustrate the advantages of the data set proposed by this work.
3. Notice that the purpose of this work is video exposure correction, so how to specifically use the complementarity between frames to assist exposure correction in the method proposed by this work?

**Suitability:**

2

---

### Official Review · Reviewer_xLXL · 2024-05-24

**Rating:** 4
**Confidence:** 2

**Summary:**

The paper addresses the problem of exposure correction in videos, particularly focusing on dynamic scenes that suffer from both underexposure and overexposure. It introduces the first real-world paired video dataset for exposure correction, named DIME (Dynamic scenes In Multiple Exposure), and proposes an end-to-end Video Exposure Correction Network (VECNet) based on Retinex theory. The dataset and the method are designed to handle the challenges posed by varying lighting conditions and motion in dynamic scenes.

**Strengths:**

- The introduction of the DIME (Dynamic scenes In Multiple Exposure) dataset is a contribution to the field of video exposure correction. This is the first real-world paired video dataset that includes both underexposed and overexposed dynamic scenes. The novelty lies in the meticulous design and execution of the data collection process, which involves synchronized dual-camera setups and precise spatial and temporal alignment.
-  By comparing VECNet with multiple existing methods, the paper provides a comprehensive evaluation.
-  The paper is well-organized, with clear sections detailing the problem, methodology, dataset creation, experimental setup, and results.

**Limitations:**

- The intricate design of the VECNet model, involving multiple sophisticated modules, can pose significant implementation challenges.
- The paper does not compare the proposed method with some of the latest advancements in unsupervised or self-supervised learning techniques for video enhancement.

**Suitability:**

2

---

### Official Review · Reviewer_VXxb · 2024-05-30

**Rating:** 2
**Confidence:** 4

**Summary:**

This paper constructs a real-world paired video dataset, which covers dynamic scenes with both underexposure and overexposure, ensuring precise spatial alignment within the dataset. To accomplish the task of video exposure correction, the paper introduces VECNet. VECNet employs a dual-stream module to separately address issues of underexposure and overexposure based on Retinex theory. Compared to other methods, this approach achieves the best performance on the proposed dataset.

**Strengths:**

1. It constructs an exposure correction video dataset with multiple exposures and precise spatial alignment.
2. It Designs an exposure correction network that performs well on the proposed dataset and demonstrates certain generalization capabilities on the LLIV-Phone dataset.

**Limitations:**

The construction of the proposed dataset needs to control the settings of different apertures and ISO parameters for the two cameras. How to ensure strictly alignment of the data needs more explanations. The noise level, the defocus phenomenon or the color tunes of the images may be varied for the different images of different settings. The effectiveness of the proposed dataset for real applications is suspicious. In additions, there are also some other concerns as following.
1. The architecture of the proposed VECNet lacks novelties. It incorporates many existing designs, such as the concepts of RCAB and Dual-stream Illumination Construction.
2. The LCDP (Local Color Distributions Prior for Image Enhancement) shares striking similarities with the Dual-stream Illumination Construction proposed in the paper. Yet the paper lacks experimental comparisons with the LCDP in the experiments. It is recommended to include the LCDP method in the comparative experiments to demonstrate the effectiveness of other architecture designs in the paper.
3. In Section 4.6 of the paper, to further demonstrate the effectiveness of the MFA module, it is suggested to conduct ablation experiments to showcase the effects of removing the MFA module in the final design.

**Suitability:**

3

---

### Official Review · Reviewer_2Df9 · 2024-06-09

**Rating:** 5
**Confidence:** 3

**Summary:**

The paper aims to solve under and over exposure problems in video, by collecting a paired real-world dataset, and training on it a novel exposure correction network.
The dataset is collected by a beam-splitter based optical system, containing incorrectly exposed videos and human touched GT.
The network structure takes advantage of Retinex theory, establishing a symmetrical structure to handle under and over exposure simultaneously.
Extensive experiment results are also provided, serving as evidence for superiority of the proposed method against existing ones.

**Strengths:**

- The symmetrical exposure correction structure, combined with alignment process in Fourier transformed domain, alleviates complexity of desired mapping to learn. This is proved by analysis and visualization in figure 13.
- Experiments provide thorough evaluation on the proposed method, from qualitative and quantitative results, to user study and ablation analysis, providing ample evidence for effectiveness of the pipeline.
- The paper is well written.

**Limitations:**

- Lack of literature review section.
- No mentioning of frame rate for videos in the dataset. Because each scene is 10 seconds, and contains 100-200 frames, it can be inferred that videos are below 20 fps. The effect of frame rate on temporal stability needs further discussion or experimentation.

**Suitability:**

3

---

### Meta-Review · Area_Chair_LrGm · 2024-07-07

**Recommendation:** Accept (Oral)
**Confidence:** 4

**Metareview:**

After checking the first round of reviews, the rebuttal from the authors and the final recommendation by the reviewers, who all lean towards the acceptance of the paper (1 weak accept and 3 borderline accept) I recommend this paper to be accepted.